# Cardiovascular Prognosis in Patients with Peripheral Artery Disease and Approach to Therapy

**DOI:** 10.3390/biomedicines11123131

**Published:** 2023-11-24

**Authors:** Antonio Curcio, Alessandra Panarello, Carmen Spaccarotella, Ciro Indolfi

**Affiliations:** 1Division of Cardiology, Department of Medical and Surgical Sciences, Magna Graecia University, 88100 Catanzaro, Italy; alessandra.panarello@studenti.unicz.it (A.P.); indolfi@unicz.it (C.I.); 2Division of Cardiology, Department of Advanced Biomedical Sciences, University of Naples Federico II, 80131 Naples, Italy; carmenannamaria.spaccarotella@unina.it

**Keywords:** limb ischemia, cardiovascular events, prognosis, revascularization, gene therapy

## Abstract

Peripheral artery disease (PAD), the pathophysiologic narrowing of the arterial blood vessels of the lower limbs due to atherosclerosis, is estimated to affect more than 200 million people worldwide and its prevalence ranges from 0.9 to 31.3% in people aged ≥50 years. It is an established marker of systemic obstructive atherosclerosis, which depicts patients at higher risk of myocardial infarction and stroke, due to the involvement of coronary and cerebral arteries in the atherosclerotic process. Therefore, identifying PAD, particularly in patients with coronary artery disease, is important to assess the cardiovascular risk score and implement specific therapies and prevention strategies. Since PAD emerged as an important clinical cardiovascular predictor, even more than other typical cardiovascular risk factors, an aggressive strategy to identify and treat PAD patients should be pursued by general practitioners, cardiologists, and vascular surgeons; similarly, preventive strategies should be implemented to improve prognosis and outcomes, particularly in patients suffering from both coronary artery disease and PAD. In this review, we describe the pathophysiology, including limb vasoconstriction after coronary angioplasty, the diagnosis of PAD, prognosis according to cardiovascular events, coronary artery disease, and heart failure. Furthermore, a large section of this review is on management, which spans from risk factors’ modification to antithrombotic therapy, and revascularization is provided. Finally, considerations about newer therapeutic options for the “desert foot” are discussed, including gene therapy.

## 1. Introduction

Peripheral artery disease (PAD) is an established marker of systemic atherosclerosis, which depicts patients at higher risk of myocardial infarction (MI) and stroke [1]. More than 200 million people worldwide suffer from PAD [2]. There are some differences in the prevalence of PAD based on the arterial territories considered [3]. In a meta-analysis, the prevalence of moderate to severe (≥50%) carotid artery stenoses was 4.2% in the whole population, and it was higher in men and in people aged 70 years or older compared to women and subjects younger than 70 years [4]. 

The prevalence of subclavian stenosis, based on the 50% sensitive method of detecting an interarm systolic blood pressure (SBP) difference >10 or 15 mmHg, is estimated to be approximately 2% in the general population, but increases to 9% in the case of concomitant lower extremities artery disease [5]. Chronic mesenteric artery disease is often underdiagnosed; in patients with atherosclerotic disease at other sites, mesenteric artery disease may be relatively common [6]. In the population of the Cardiovascular Health Study, renal artery stenosis ≥60% was detected in 9.1% of men and 5.5% of women [7]. Lower extremities artery disease usually appears after the age of 50 years, with an exponential increase after the age of 65 years [1,7]. This rate reaches roughly 20% by the age of 80 years. In most studies, the proportion of symptomatic lower limb disease is 1:3 to 1:5 of all affected patients (Figure 1) [8]. 

The development and progression of PAD are associated with traditional cardiovascular (CV) risk factors (Table 1). Both past and current smoking habits have been associated with PAD and the risk increases with smoking intensity [9]. The high prevalence of hypertension, particularly among older patients, makes it a significant contributor to the total burden of PAD in the population [2,4,7,10]. Hypercholesterolemia demonstrated to be a strong and independent risk factor for clinical PAD. HDL-cholesterol has been shown to be protective [8,9,10]. Diabetes mellitus (DM) is strongly associated with PAD and the risk increases with diabetes duration [11]. Moreover, diabetic patients with concomitant PAD have attributed worse outcomes than non-diabetics [8,9,10,11].

Identifying PAD, particularly in patients with coronary artery disease (CAD), is important to assess the CV risk score and implement a prevention strategy [8,9,10,12,13,14,15]. 

The aim of this review will be to describe the pathophysiology, including limb vasoconstriction after coronary angioplasty, the diagnosis of PAD, prognosis according to CV events, CAD, and heart failure. Furthermore, a large section of this review will be on management, which spans from risk factors’ modification to antithrombotic therapy, and revascularization is provided. Finally, considerations about newer therapeutic options for the “desert foot”, including gene therapy, are discussed.

## 2. Pathophysiology of PAD

Endothelial dysfunction is a systemic disorder that represents the early stage of atherosclerosis and the initial step in the origin and development of atherosclerotic CV diseases [16].

PAD patients have a decreased nitric oxide (NO)-mediated vasodilatation in response to shear stress induced by an increase in blood flow and hormonal agonists like acetylcholine, serotonin (5-HT), thrombin, and bradykinin (Table 2) [17]. This condition leads to an impaired oxygen (O_2_) delivery capacity and an altered O_2_ extraction in peripheral vessels, thus there is a supply–demand mismatch. 

The arterial blood supply is regulated by perfusion pressure and vascular resistance. If stenoses are present, blood flow is reduced proportionally to the narrowing severity [18].

In patients suffering from intermittent claudication, there are single or multiple occlusive lesions that usually permit a normal arterial blood flow at rest and limit the blood flow at the muscular level during exercise (Figure 2). In the case of critical limb ischemia, there are multiple occlusive lesions, thus the metabolic needs of the muscle outstrip the supply of O_2_ and nutrients even in resting condition, causing rest pain and tissue loss. In the case of acute limb ischemia, a thrombotic process joins the pathophysiology process, leading to total arterial occlusion [16,17,18].

Evidence from an electrophysiologic examination showed that there is a partial axonal denervation of skeletal muscle which involves only type II glycolytic fast-twitch fibbers and leads to decreased muscle strength and a reduced exercise capacity (Table 3) [19].

The muscle distal to the atherosclerotic lesion shifts from an aerobic to anaerobic metabolism with the accumulation of lactate and other metabolites (adenosine, NO, potassium, and a hydrogen ion) followed by the activation of local sensory receptors with ischemic pain and intermittent claudication during the exercise [20].

Across the stenosis, the blood pressure gradient increases in a nonlinear way, highlighting the severity of stenosis in the case of high blood flow rates. In case of 50% stenosis or more, a blood pressure gradient is present even at rest. Moreover, impairments in vasomotor reactivity and microcirculation interfere with the normal blood flow since PAD patients have a decreased vasodilator capability in response to biochemical stimuli and a reduction in perfused skin capillaries. The vasomotor reactivity of the limb has been studied in conjunction with the angioplasty of the coronary arteries; in fact, after angioplasty, distal coronary vasoconstriction mediated by α1- and α2-adrenergic receptors occurs, and a subsequent sympathetic overdrive caused by balloon dilatation leads to a vasoactive substance release from the platelets. Accordingly, a decrease in the forearm blood flow and an increase in forearm vascular resistance was observed when peripheral circulation was assessed before and after coronary angioplasty during the intra-arterial continuous infusion of a calcium-antagonist or α-adrenergic blockade agent (verapamil or phentolamine, respectively), thus demonstrating a link between systemic vessels and coronary artery dilation [21].

From the start of the atherosclerotic process, there are several steps that lead to plaque formation. The American Heart Association (AHA) classifies atherosclerotic lesions as follows: type I lesions, in which there is an increase of macrophages and foam cells; type II lesions consist primarily of foam and muscle cells covered by lipids, also known as “fatty streaks”; in lesions classified as type III, sporadic aggregations of extracellular lipid droplets and particles are observed; type IV lesions are larger, convergent, and characterized by a more unstable core of extracellular lipids that prelude to fibroadipose plaques; type V lesions characterize patients ≥40 years old, with a lipid core that may also contain thick layers of fibrous connective tissue; type Vb refers to calcified lesions, while lesions containing fibrous connective tissue with little or no accumulated lipid or calcium are classified as type Vc; and, finally, type VI occurs when either a fissure, hematoma, or thrombus complicate the lesion [9,19,20].

A recent study found that patients with acute coronary syndrome (ACS) and concomitant PAD have a higher risk of major adverse cardiovascular events (MACE), limb ischemic events, and CV mortality, and this is partially due to increased inflammation [12].

## 3. Diagnosis of PAD

Accurate clinical history and physical examinations are key steps in PAD diagnosis and management. Personal and family clinical history should be investigated for CAD, cerebrovascular disease, aortic aneurysm, and PAD (Figure 2). In addition, it is necessary to assess the clinical status of the patient, becoming informed about typical or atypical symptoms through standardized questionnaires. Signs suggestive of arterial ischemia such as pallor, atrophy or ischemic ulcers, auscultation of arterial bruits, and the palpation of reduced distal arterial pulses should be investigated for both upper and lower extremities, therefore concluding the classification according to the Fontaine or Rutherford scales [9,10]. 

Patients with signs or symptoms suggestive of PAD and asymptomatic patients with risk factors should be non-invasively tested to confirm the diagnosis and to characterize the severity of the disease through a segmental pressure measurement and ankle-brachial index (ABI). Both tests are based on the same principle: some pneumatic cuffs are placed on different portions of the upper and lower extremities for measuring SBP. The segmental pressure measurement detects arterial stenosis when blood pressure gradients in excess between consecutive cuffs of 20 mmHg in the lower extremities, and of 10 mmHg in the upper extremities are found. The ABI is a simplification of the segmental pressure measurement, based on the ratio of SBP at the ankle and at the brachial arteries. According to the European Society of Cardiology (ESC) guidelines, normal ABI values range from 1.00 to 1.40, a value of 0.91 to 0.99 is an ABI borderline, while an ABI of 0.90 or less is abnormal and is associated with a two- or three-fold increased risk of total and CV death. An ABI greater than 1.40 identifies arterial stiffening due to arterial calcification, typical of DM or chronic kidney disease (CKD), and is associated with a higher risk of CV events and death. The ABI test can also help in assessing the severity of PAD since a range from 0.5 to 0.8 is often related to intermittent claudication, while an ABI value of 0.49 or less is suggestive for critical limb ischemia [8].

Treadmill exercise testing can be useful to better define the patient walking capacity and claudication onset time [18]. Normally, the blood pressure increases simultaneously in the upper and lower extremities, thus the ABI value is constantly 1.0 or more. In the case of PAD, larger decreases are observed after exercise. 

Pulse volume recording provides a graphic illustration of the beat-to-beat volumetric change; in the case of distal vascular stenosis, the pulse contour does not show the dicrotic notch and has a slower downslope, while the pulse wave is totally absent in the case of critical limb ischemia. 

Duplex ultrasound includes B-mode echography, pulsed-wave, continuous, color, and power Doppler. It allows the clinician to detect, localize, and quantify the stenoses.

Other imaging methods are used to better characterize the lesion and the arterial anatomy, often in a setting of pre-revascularization analyses, such as computed tomography angiography (CTA), magnetic resonance angiography (MRA), and contrast-enhanced angiography. 

While CTA has a high resolution and feasibility for patients previously treated with paramagnetic prosthesis, MRA does not require ionizing radiation and is more suitable for CKD patients or in case of allergy [8,9,10].

## 4. Prognosis of PAD

### 4.1. Cardiovascular Events

Impaired vasodilation and endothelial dysfunction have been shown to predict CV events, even in patients with angiographically normal coronary vessels (Table 3). 

By starting from a univariate analysis of 84 patients through an Doppler echocardiography of carotid arteries, it has been shown that patients with CV and cerebrovascular events had a significantly older age and there was a higher prevalence of male sex and baseline carotid lesions; in fact, the multivariate analysis found that baseline carotid lesions and flow-mediated dilatation below the median value were the sole predictive variables. Additional studies demonstrated that carotid lesions are strong independent predictors and increase the risk of future CV events through the evaluation of systemic vasoreactivity and carotid intima/media thickness [22].

On the other hand, endothelial dysfunction detected by non-invasive peripheral arterial tonometry has been similarly included among the predictors of late adverse CV events since the natural logarithmic scaled reactive hyperemia index less than 0.4 was associated with a 31% occurrence of hospitalization, MI, revascularization, and cardiac death in a seven-year follow-up [23].

Interestingly, in a recent meta-analysis, the parameters discussed above were considered together, thus suggesting that flow-mediated dilation may be more useful as a screening test for recurrent CV events in patients at high risk compared to the general population [24].

### 4.2. Coronary Artery Disease

Patients with PAD often have comorbidities such as systemic hypertension, DM, family history of CAD, history of MI, and surgical or percutaneous revascularization whose likelihood of three-vessel coronary disease is even higher. While several studies investigated the association between CAD and PAD and their impairment of life expectancy and prognosis, it should also be considered that preoperative CAD screening is mandatory for elective carotid endarterectomy.

A study including 14,000 patients found that at any time during follow-up, the group of patients with a vascular disease had a 25% increased risk of death, after controlling for all other CV risk factors; interestingly, among the main predictors, cerebrovascular disease alone was correlated with a worse prognosis [14].

An investigational study on the prognostic role of brachial artery reactivity in patients suffering from CAD who had undergone percutaneous coronary revascularization (PCI) showed that the increased risk of future CV events is related to endothelial damage and an abnormal vasomotor function [13]. Moreover, this study reported the independent prognostic value of flow-mediated dilation in acute coronary syndromes (ACSs) patients, highlighting how flow-mediated dilation might be useful in assessing the progression of myocardial damage after PCI. 

In the same lane, another study addressed the ACSs and concomitant history of PAD in addition to signs of diffuse and persistent humoral inflammation identified by high levels of hs-CRP and soluble fms, like tyrosine kinase-1 (sFlt-1), associating them with an increased rate of major limb ischemia, adverse CV events, and CV death [12].

Since PAD emerged as being an important clinical predictor, even more than other typical CV risk factors such as previous MI or angina, a particularly aggressive medical strategy aimed at secondary prevention should be used in patients suffering from both CAD and PAD to ameliorate their prognosis.

### 4.3. Heart Failure

As for ACS, PAD has been associated also with chronic coronary syndromes as well as its progression through the heart failure (HF) of an ischemic aetiology. In fact, New York Heart Association IV functional class HF patients with concomitant endothelial dysfunction have an increased mortality risk. 

The contribution by endothelial dysfunction to HF progression has been variably linked to abnormal myocardial perfusion, myocardial damage, and impairment in the exercise-induced release of NO, with an overall deterioration of tissue perfusion that may contribute to a reduced exercise capacity in HF patients. Moreover, HF patients seem to have a reduction of both endothelium-dependent and -independent vasodilatation, suggesting that probably there are some vascular structural abnormalities which further contribute to the pathogenesis and progression of HF. 

A study with 82 patients explored the possible association between brachial artery endothelial function and mortality risk in advanced NYHA class IV ischemic patients with a mean left ventricular ejection fraction (EF) of 22 ± 3%. It demonstrated that in patients with a flow-mediated dilation lower than or equal to the median, hospitalizations for HF exacerbations were twice as high as those of patients with a flow-mediated dilation above the median, with added mortality [25]. 

Conversely, other studies have been conducted to investigate the possible association of endothelial dysfunction and HF with a preserved EF. In more detail, a prospective cohort study and a non-randomized retrospective study both showed that a reactive hyperemia index below 0.49 significantly correlated with adverse CV events in the HFpEF setting [26,27].

## 5. Management of PAD

### 5.1. Risk Factor Modification

Strategies for risk factor modification represent the first step in the management of PAD. 

Since several pieces of evidence showed that cigarette smoking increases the atherosclerotic risk, and PAD patients who smoke have a greater risk of acute CV events and mortality than non-smokers, smoking cessation must be one of the first goals to be gained in the prevention and treatment of PAD (Figure 3). In fact, both the ESC and the AHA guidelines recommend assisting PAD patients who smoke in developing a plan for quitting tobacco addiction, even with pharmacological support based on bupropion, varenicline, and nicotine replacement [8,9].

Another goal to achieve is health status, through a low-calorie diet and pharmacotherapy including statins, omega-3 polyenoic fatty acids, ezetimibe, PCSK9 inhibitors, siRNA, or bempedoic acid when appropriate. According to ESC guidelines, all patients diagnosed with PAD should receive lipid-lowering therapy and gain an LDL target of <70 mg/dL or decrease by ≥50% in the case of initial LDL levels between 70 and 135 mg/dL [8]. Moreover, in these patients, strict glycemic control and DM management through target-specific therapies such as sGLT2 inhibitors and GLP-1 receptor agonists are recommended for reducing the risk of MACE and improving CV outcomes in patients with atherosclerosis, including PAD.

### 5.2. Exercise Therapy

In patients with PAD suffering from intermittent claudication, structured exercise therapy can improve quality of life and increase maximal walking distance. Supervised exercise therapy should be preferred over unsupervised [8,9,10,18] and consists of walking to the maximal distance, cycling, upper-arm ergometry, and strength training (Figure 3). In case of a reduction in severe daily life activities, exercise therapy should be performed before and in association with revascularization [8].

### 5.3. Antithrombotic Therapy

The antithrombotic regimen based on acetylsalicylic acid, statins, thienopyridines, and angiotensin-converting enzyme inhibitors depends on symptoms, atherosclerosis localization, and a history of revascularization [8,15]. Antiplatelet therapy (APT) is recommended in all patients with symptomatic PAD (Figure 3). 

Long-term single APT is mandatory in all patients with carotid artery stenosis, and it has a class I recommendation in case of symptomatic stenosis and class IIa recommendation in case of asymptomatic >50% stenosis with a low bleeding risk.

In patients suffering from lower extremities disease, long-term single APT is recommended only if symptomatic or after revascularization. For single APT, clopidogrel may be preferred over aspirin. 

Atrial fibrillation (AF) and PAD in elderly patients increase rates of stroke, amputation, and death; moreover, PAD is more severe and related to in-hospital complications in AF patients.

AF management through chronic anticoagulation therapy alone in concomitant PAD is recommended only in a CHA_2_DS_2_VASc score ≥ 2. At the same time, a combination with a single APT (acetylsalicylic acid or clopidogrel) for at least 1 month is suggested in the case of recent revascularization and when the bleeding risk is low. On the other hand, vitamin K antagonist administration in patients with stable CV disease showed a reduction in the risk of CV events, while in PAD patients, no benefits were observed at the cost of a significantly higher rate of bleeding, including intracranial bleeding. 

More importantly, a recent multicentric international trial compared three APTs in patients with stable atherosclerotic vascular disease, including a reduced (2.5 mg) dose of rivaroxaban twice daily, demonstrating a significant cut in mortality and the risk of major adverse CV events when rivaroxaban was added to acetyl salicylic acid [28]. Therefore, other international CV societies [10], even considering other comorbidities [29], recently joined the recommendation of 2.5 mg of rivaroxaban twice daily plus acetylsalicylic acid as an alternative antithrombotic regimen in patients with chronic atherosclerotic disease who have a low bleeding risk and are at high risk for ischemic events [30].

## 6. Revascularization and Outcomes

Endovascular treatment is recommended as the first strategy for revascularization in short aorto-iliac and femoropopliteal occlusive lesions (Figure 3) [8]. Conversely, surgical revascularization is recommended in the case of infra-popliteal disease in patients suffering from chronic limb-threatening ischemia [31]. In PAD patients with long lesions, when the surgical risk is not too high, the need for bypass surgery depends on the lesion’s localization and the availability of an autologous vein. In the setting of intermittent claudication, the first strategy is medical therapy in association with exercise, while revascularization should be considered only in case of daily life limitation.

Differential referral to revascularization approaches has been addressed by a recent trial showing that in patients suffering from severe limb ischemia due to infrainguinal atherosclerosis, who seem suitable for both percutaneous and surgical treatment, the choice must be guided by individual characteristics and local expertise [32]; however, despite the high failure and re-intervention rate associated with angioplasty, the percutaneous approach should be preferred in patients with severe comorbidities and a life expectancy of less than 1–2 years, leaving the surgical option as a bail-out strategy [33]. In contrast, in patients who have a life expectancy of more than two years and are considered at low surgical risk, it seems more appropriate to perform surgery as a first approach, despite the increased morbidity and costs, because of its apparent improved durability and successful rate.

A risk score developed for PAD considers the 30-days postoperative mortality and major lower-limb amputation in patients with three comorbidities among DM, CAD, foot gangrene, or urgent surgery [34]. Moreover, the “PIII” risk score addressed a few variables at the initial presentation like dialysis dependency, tissue loss, advanced age, a low hematocrit, and advanced CAD, hence stratifying patients with PAD and surgically correctable infrainguinal disease into three categories of expected amputation-free survival [35].

On the other hand, several studies investigated endovascular treatment outcomes in patients with PAD. A retrospective multicentric study demonstrated increased mortality for pneumonia and sepsis in patients with critical limb ischemia compared to patients with claudication [36].

Such findings were confirmed in terms of MACE [37], with the highest risk of amputation being represented by subjects affected by critical limb ischemia and DM after endovascular surgery [38].

## 7. Future Scenarios

### 7.1. No-Option Critical Limb Ischemia

The failure in blood flow improvement in spite of surgical and/or percutaneous intervention qualifies the patient for amputation. Recently, the issue of the “desert foot” has been addressed by several clinical trials testing the experimental administration of different molecules (Figure 3), which are pushing biotechnology industries and regulatory agencies to release such therapies in the next few years [39]. Current approaches for delivering a genetic material that facilitates angiogenesis and/or arteriogenesis include a lentivirus, adenovirus, and adeno-associated virus as well as nonviral plasmid DNA [40,41].

An intramuscular injection of VEGF (vascular endothelial growth factor) represents the first gene-therapy-based approach for “no-option critical limb ischemia” [42], followed by the implementation of hepatocyte growth factor (HGF) [43]. On the other side, alternative routes of administration have been considered for L-carnosine [44] to stabilize HIF-1α (hypoxia-inducible factor-1α) and promote angiogenesis when given orally, as well as subcutaneous liraglutide (a glucagon-like peptide 1 receptor agonist) to promote peripheral perfusion (Figure 4) [45].

### 7.2. Remote Ischemic Conditioning

Peripheral ischemia has been associated with clinical neurological benefits. In contrast with intravenous thrombolysis and a thrombectomy, which are the standardized reperfusion therapies for acute ischemic stroke, several experimental studies have addressed the effect of remote ischemic conditioning, which is represented by transient cycles of upper extremities or lower extremities ischemia and reperfusion [46]. This method induces the release of humoral factors and the activation of the nervous system in order to support and stimulate brain cell viability and has proven to reduce the infarct size. The RICAMIS trial, which investigated the efficacy of the bilateral upper extremities application of remote ischemic conditioning within 48 h after symptoms onset up to 10–14 days later, demonstrated that patients treated with remote ischemia had a greater neurological outcome at 90 days compared to those who were not randomized for this adjunctive treatment [47]. Recently, remote ischemic conditioning was applied unilaterally on the upper extremities since the first medical contact and for the next 7 days in patients with a diagnosis of acute stroke, without observing any functional amelioration at 90 days compared to those treated with usual care only [48]. While it has been postulated that assessments of the hyperacute phase in the RESIST trial have led to such conflicting results, more evidence about the clinical efficacy of this technique might dramatically change the therapeutic routine.

## 8. Conclusions

Patients with PAD have an increased risk for adverse CV events as well as the risk of limb loss and an impaired quality of life. Patients with PAD frequently have concomitant CAD and cerebrovascular diseases. The traditional indications for arterial angioplasty in the management of PAD are ischemic rest pain, ischemic ulceration, or gangrene. However, severe claudication that prevents minimal ambulation and infra-popliteal angioplasty has been advocated by some interventionalists. While current revascularization strategies aimed at limb salvage are fundamental for ameliorating the outcome of PAD and related CV diseases, risk factor modification should always be pursued. In a hypothetically near future, major biotechnological achievements will provide newer vectors targeting molecular and cellular mediators that eventually promote the enlargement of pre-existing arteriolar connections as collateral channels through arteriogenesis. Addressing such innovations, alongside traditional proangiogenic approaches, in preclinical studies of elderly animal models affected by multiple risk factors, and hopefully in large randomized, double-blind, controlled clinical trials, is warranted.

## Figures and Tables

**Figure 1 biomedicines-11-03131-f001:**
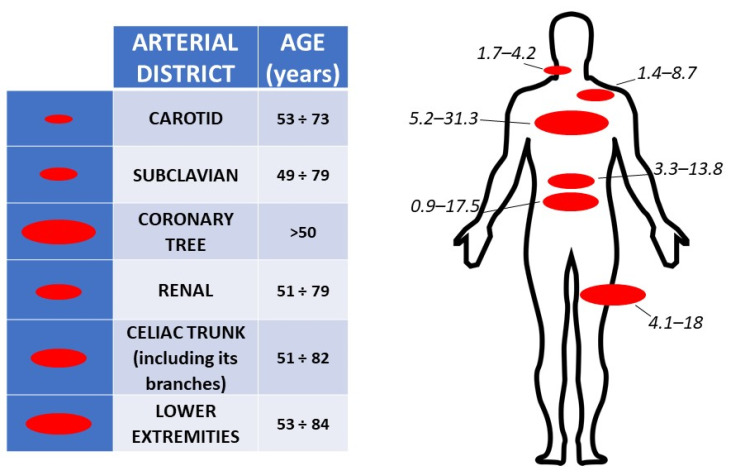
Likelihood of thrombosis in different arterial vessels. Involvement of arterial conduits by atherosclerotic disease is quite variable and begins in the first years of life, reaching its clinical manifestations as indicated in the age ranges reported on the right column of the left panel; the magnitude of atherosclerosis is represented by oval shapes of different areas (left column/left panel) which are reported for more clarity upon the schematic human body of the right panel. Numbers in italics indicate the prevalence of the disease for each arterial district.

**Figure 2 biomedicines-11-03131-f002:**
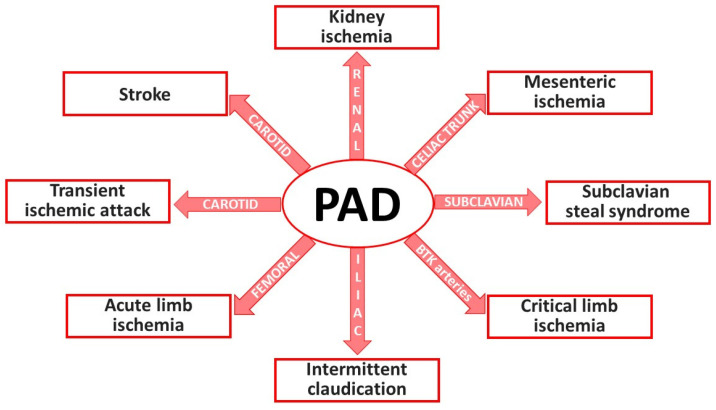
Clinical manifestations of PAD according to the involved arterial district. Main PAD-related pathologies are depicted in rectangles, while arrows report the arterial vessels that are affected by atherothrombosis mechanisms. Abbreviations: BTK, below the knee.

**Figure 3 biomedicines-11-03131-f003:**
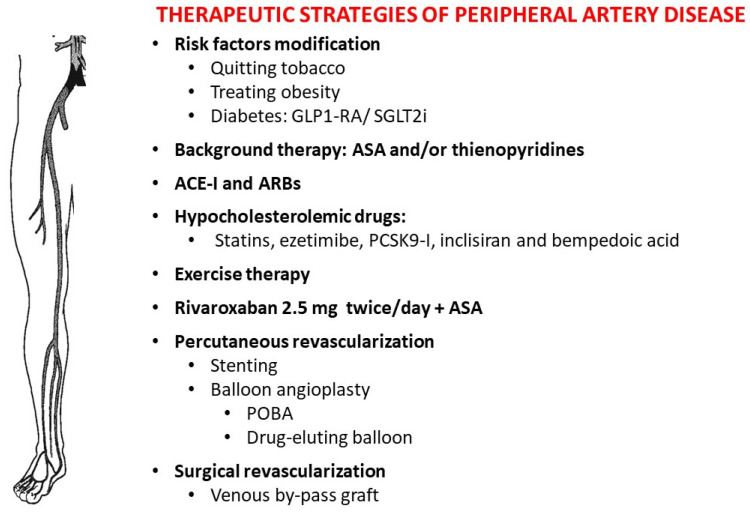
Established therapy for PAD. Current therapeutic strategies consider lifestyle modifications, lipid-lowering, antithrombotic drugs, and percutaneous and surgical revascularizations. Abbreviations: GLP1-RA, glucagon-like peptide 1-receptor agonist; SGLT2i, sodium–glucose co-transporter-2 inhibitors; ASA, acetil salycilic acid; ACE-I, angiotensin converting enzyme inhibitors; ARB, angiotensin II receptor blocker; PCSK9-I, proprotein convertase subtilisin/kexin type 9 inhibitors; POBA, plain old balloon angioplasty.

**Figure 4 biomedicines-11-03131-f004:**
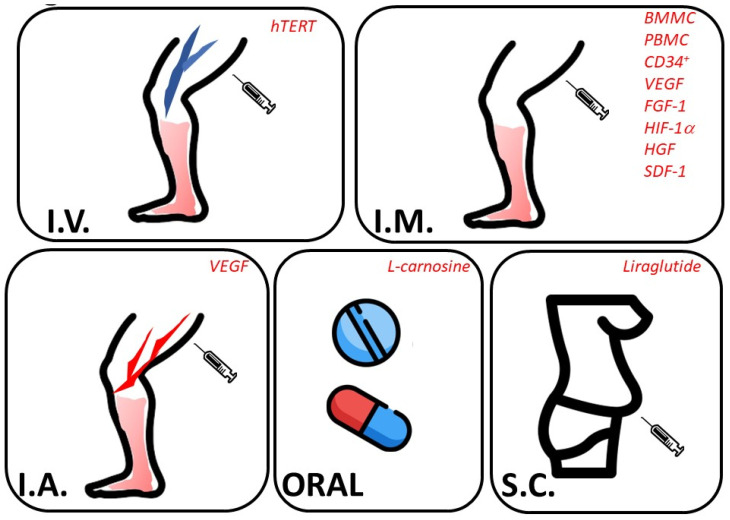
Promising alternative therapies for PAD. The figure reports the routes of administration (capital letters, bold) and the emerging non-pharmacologic approaches together with new indications for drugs (italics, red color) that are currently under investigation in clinical trials aimed at reducing amputation risk. Abbreviations: hTERT, human telomerase reverse transcriptase; BMMC, bone marrow mononuclear cells; PBMC, peripheral blood mononuclear cells; VEGF, vascular endothelial growth factor; FGF-1, fibroblast growth factor-1; HIF-1α, hypoxia-inducible factor-1 alpha; HGF, hepatocyte growth factor; SDF-1, stromal cell-derived factor-1.

**Table 1 biomedicines-11-03131-t001:** Conventional risk factors for PAD.

Lifestyle	Comorbidities	Genetic Background
Western diet	Systemic hypertension	Race
Cigarette smoking	Dyslipidemia (including all atherogenic lipid subsets or FH)	Ethnicity
Sedentary behavior	Diabetes mellitus	CHIP
Alcohol consumption	Obesity	
	Homocysteine	
	High-sensitive C-reactive	
	Fibrinogen	
	Chronic kidney disease	

List of abbreviations: FH, familial hypercholesterolemia; CHIP, clonal hematopoiesis of indeterminate potential.

**Table 2 biomedicines-11-03131-t002:** Molecular and cellular determinants of PAD.

Molecules	Cellular Effects
hs-CRP	Endothelial dysfunction
α-defensins	Inhibition of the fibrinolytic system Stimulation of LPa and LDL binding to the extracellular matrix of ECs Endothelial dysfunction
sFlt1	Endothelial dysfunction
Fibrinogen	Fibrin formation
	Blood viscosity
	Platelet aggregation
	Vascular ECs and SMCs proliferation
FVII	Activation of coagulation Correlation with hypertriglyceridemia
FXIII	Stabilization of the fibrin clot
FV Leiden	Fibrin clot formation
FII/Prothrombin	Fibrin clot formation
Decreased thrombomodulin plasma levels	Deceleration of the thrombin-induced activation of protein C
GP IIb/IIIa	Adhesion of platelets to exposed subendothelial extracellular matrix Platelets mutual interaction Binding of vWF
GP Ia/IIa	Platelet adhesion to the exposed vessel wall
GP Ib/IX/V	Binding of vWF
t-PA	Endothelium-dependent activation of the fibrinolytic system
PAI-1	Inhibition of t-PA Correlation with hypertriglyceridemia
TAFI	Regulation of fibrinolysis Prolongation of clot lysis time
Hyperhomocysteinemia	Endothelial dysfunction
Decreased NO levels	Impaired SMCs relaxation and vascular tone regulation Vascular SMCs proliferation Leukocyte adhesion to the endothelium Reduced ROS elimination Adhesion, activation, and aggregation of platelets
PON1 deficiency	Loss of HDL protective function Induction of monocyte–endothelial interaction
Plasma GPx-3 deficiency	Reduction of bioavailable NO Impaired metabolism of ROS
Connexin 37	Atherosclerotic plaque formation

Legend: main categories are indicated in bold letters; subcategories are indicated in italics. List of abbreviations: hs-CRP, high-sensitivity C reactive protein; LPa, lipoprotein-a; LDL, low-density lipoproteins; ECs, endothelial cells; sFlt1, soluble fms-like tyrosine kinase 1; SMCs, smooth muscle cells; FVII, factor VII; FXIII, factor XIII; FV Leiden, factor V Leiden; FII, factor II; GP, glycoprotein; vWF, von Willebrand factor; t-PA, tissue-type plasminogen activator; PAI-1, plasminogen activator inhibitor-1; TAFI, thrombin-activatable fibrinolysis inhibitor; NO, nitric oxide; ROS, reactive oxygen species; PON1, paraoxonase-1; HDL, high-density lipoprotein; GPx-3, glutathione peroxidase.

**Table 3 biomedicines-11-03131-t003:** Pathophysiological substrates of PAD.

Regulation of Blood Supply to Limb	Structure and Function of the Skeletal Muscle	Atherosclerotic Plaque Vulnerability and Disruption
Flow-limiting lesion (severity of arterial vascular stenosis and inadequate collateral vessel formation)	Axonal denervation of the skeletal muscle	Chronic low-grade inflammation (IL-6, cytokines, hs-CRP, and α-defensins)
Impaired vasodilation (decreased NO and reduced responsiveness to vasodilators)	Loss of type II glycolytic fast-twitch fibers	
Accentuated vasoconstriction (TX, 5-HT, ATII, ET, and NE)	Impaired mitochondrial enzymatic activity	
Abnormal rheology (reduced red blood cell deformability, increased leukocyte adhesivity, platelet aggregation, microthrombosis, and increased fibrinogen)		
Altered microcirculation (decreased number of perfused skin capillaries)		

Legend: main substrates are indicated in bold letters; subcategories are indicated in italics. List of abbreviations: NO, nitric oxide; TX, thromboxane; 5-HT, 5-hydroxytryptamine; ATII, angiotensin II; ET, endothelin; NE, norepinephrine; IL-6, interleukin-6; hs-CRP, high-sensitivity C reactive protein.

## Data Availability

Not applicable.

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
