# Peer review of "Cardiovascular Prognosis in Patients with Peripheral Artery Disease and Approach to Therapy"

_biomedicines, 2023, doi:10.3390/biomedicines11123131_

Round 1

Reviewer 1 Report

Comments and Suggestions for Authors

General commets

Thank you for the opportunity to review this work wich aimed to  describe the pathophysiology, including limb vaso constriction after coronary angioplasty, diagnosis of PAD, prognosis according to CV events, CAD, and heart failure; furthermore, a large section on management, which spans from risk factors modification toward antithrombotic therapy and revascularization is provided, and finally, considerations about newer therapeutic options for the “desert foot”, including gene therapy, are discussed.

The manuscript is well writen. I have only a few suggestions  for the authors to improve the manuscript.

Specific comments 

Line 168 - write in full the acronym BP -  blood pressure

Line - 246 - New York Heart Association (NYHA)

Line - 252 - NO - Nitric Oxide 

Reviewer 2 Report

Comments and Suggestions for Authors

The article has defined the goal to address Peripheral Arterial Disease (PAD)'s cardiovascular prognosis and therapeutic approach. The authors need to make several major revisions. The definition for PAD needs to be clarified. PAD is related mainly to lower limbs arteries. However, authors tend to include in the PAD all the extra-coronary sites of atherosclerosis. The article spends much time on epidemiology, risk factors, pathophysiology or diagnosis, like in a manual. Prognosis and therapeutic approaches are presented in summary. The area is so vast, that it is very difficult for authors to cover all the relevant information.

Reviewer 3 Report

Comments and Suggestions for Authors

This review paper dedicated to peripheral artery disease and includes pathophysiology diagnosing, prognosis; such treatment as risk factors modification, antithrombotic and lipid-lowering therapy, revascularization, gene therapy.

The paper is well written and illustrated, the references are comprehensive. There are no any issues regarding this paper.

Round 2

Reviewer 2 Report

Comments and Suggestions for Authors

The revised article keeps the structure of a broad overview of PAD (pathophysiology, diagnosis, prognosis and treatment), covering much more than the title of the article suggests.